# Strategies to maintain recovery from alcohol problems during the COVID-19 pandemic: Insights from a mixed-methods national survey of adults in the United States

**Paul A. Gilbert** [1]*, **Loulwa Soweid**[1], **Paul J. Holdefer**[1], **Sarah Kersten**[1], **Nina Mulia**[2]

**1** Department of Community and Behavioral Health, University of Iowa, Iowa City, Iowa, United States of America, **2** Alcohol Research Group, Public Health Institute, Emeryville, California, Unites States of America

* paul-gilbert@uiowa.edu

**Data Availability Statement:** As the data set includes verbatim qualitative responses from study participants, it may contain potentially identifying

## Abstract

The COVID-19 pandemic has been associated with poorer mental health and, in some cases, increased alcohol consumption; however, little is known about the pandemic's effects on people in recovery from alcohol use disorder (AUD), especially how they have coped with novel stressors. Our mixed-methods study investigated strategies used to maintain recovery during the pandemic, with attention to variation by gender. We analyzed data obtained in fall 2020 from an online US national survey of adults with resolved AUD (n = 1,492) recruited from KnowledgePanel, a probability-based cohort of non-institutionalized adults maintained by Ipsos for internet-based research. Participants endorsed possible coping strategies on a 19-item choose-all-that-apply list, which were analyzed using chi-square tests. In addition, 1,008 participants provided text responses to an open-ended question about their strategies to maintain recovery during the pandemic, which were coded and analyzed using an inductive, thematic approach. The majority of our sample met criteria for severe lifetime AUD (72.9%), reported being in recovery more than five years (75.5%), and had never used specialty AUD services or mutual-help groups (59.7%). The ordering of the coping strategies was quite similar for women and men; however, the top strategy (talking with family and friends by phone, text, or video) was endorsed more frequently by women than men (49.7% vs. 36.1%; p < .001). Among qualitative themes, "staying connected" was the most common. It was dominated by statements about family, with women mentioning children more often than men. Among other themes, "cognitive strategies" mirrored established therapeutic modalities, and "active pursuits" aligned with many recent recommendations for service providers working with substance-using populations during the pandemic. A minority of participants invoked "willpower" for recovery or stated that pandemic restrictions helped by reducing exposure to relapse risks. These findings shed light on recovery mechanisms during the COVID-19 pandemic and suggest potential intervention targets to support recovery during other catastrophic events, such as natural disasters.

or sensitive information. A limited-use data set is available under a Data Use Agreement with the University of Iowa. Please submit all data use request to them via email (tim-shie@uiowa.edu).

**Funding:** This study was supported by the National Institutes of Health (R01AA027266). The funders had no role in study design, data collection and analysis, decision to publish, or preparation of the manuscript. The content is solely the responsibility of the authors and does not necessarily represent the official views of the National Institutes of Health.

**Competing interests:** The authors have declared that no competing interests exist.

## Introduction

The COVID-19 pandemic was a source of tremendous stress with impacts on mental and behavioral health. Stay-at-home restrictions and social distancing behaviors were associated with worsening symptoms of depression, anxiety, and acute stress [1–3]. Indeed, several studies have reported higher prevalence of anxiety or depression during the COVID-19 pandemic compared to earlier periods [4–6], and one study's longitudinal results showed that already elevated depressive symptoms increased further from 2020 to 2021 [7]. Notably, Czeisler and colleagues found that 41% of a representative panel of United States (US) adults endorsed at least one symptom of anxiety, depressive, or trauma and stress-related disorders with regards to the pandemic, and reported having begun or increased substance use as a coping mechanism [8]. There are indications that the prevalence of anxiety and depression symptoms may be stabilizing or decreasing after initial peaks at the start of the COVID-19 pandemic [9, 10]; however, the ultimate resolution of mental and behavioral health concerns due to the pandemic remains unknown.

Importantly, the mental health impacts of the pandemic could affect recovery from alcohol use disorder (AUD). Stress and its relationship to both alcohol consumption and AUD has long been posited by theoretical frameworks and supported by empirical findings [11, 12]. For example, exposure to stress has been associated with an increase in alcohol consumption as a coping strategy in individuals with a history of alcohol use disorder [13, 14]. In addition, large-scale disasters have been studied as stressors that affect subsequent alcohol use. Boscarino and colleagues observed that New York City residents with greater exposure to the September 11 attacks reported greater alcohol consumption up to two years afterwards [15]. Furthermore, individuals who have experienced major natural disasters such as Hurricane Katrina and Hurricane Sandy demonstrated greater alcohol consumption and binge drinking behaviors following those events [16–19].

Evidence on alcohol consumption during the COVID-19 pandemic has been mixed, showing both increased and decreased consumption [20]. Among studies in the US, Avery and colleagues observed that 11% of their sample of adult twins reported drinking less during the early stages of the pandemic [21]. However, Grossman and colleagues found that adults in their general population sample who reported increased COVID-19 pandemic related stress also indicated greater alcohol consumption [22]. Similarly, Killgore and colleagues' national general population survey found that Alcohol Use Disorder Identification Test (AUDIT) scores increased every month for individuals who were under lockdown orders or stay-at-home restrictions, but did not increase for those not under such restrictions [23]. Among European studies, Kim and colleagues found that 24% of their sample of individuals with pre-existing AUD had increased their alcohol intake since the beginning of lockdown, and that nearly one in five (17%) previously abstinent patients relapsed during the lockdown [24]. Furthermore, Yazdi and colleagues found that patients with AUD who had either relapsed during the COVID-19 pandemic or had continued to drink alcohol since before the pandemic reported greater alcohol cravings and consumption as compared to those who had remained abstinent [25]. In addition, two multi-national review studies have reported mixed findings, with some segments of the population decreasing drinking and other segments increasing drinking [26, 27].

Gender differences in recovery warrant consideration, especially as some research has found that women experienced greater mental health impacts than men during the COVID-19 pandemic, which in turn may affect alcohol use [28–30]. One study found that women who scored higher on COVID-related anxiety were more likely to indicate an increase in daily and binge drinking, and start drinking earlier during the day than peers who scored lower [31]. In

addition, two studies conducted early in the COVID-19 pandemic found that women in the US significantly increased the frequency of their drinking and amount of alcohol they consumed while men did not show comparable increases [32, 33]. To our knowledge, there have been no published studies of COVID-related stressors and relapse risk by gender; however, previous studies have found that women with AUD are at a greater risk of relapse than men when experiencing negative affect and interpersonal stress [34–36].

Despite growing evidence that the COVID-19 pandemic has been associated with poorer mental health outcomes and (in some cases) increased alcohol consumption, there has been scant attention to its impact on people in recovery from AUD, especially factors that supported recovery during this stressful period. To extend our understanding of recovery maintenance, we conducted a mixed-methods investigation of the experiences of people with resolved AUD. Given that the COVID-19 pandemic presented novel stressors and that coping strategies and recovery from AUD are complex phenomena, we believed that drawing upon quantitative and qualitative data would allow us to more fully address our research aims than either approach alone. Specifically, we wanted to describe the range of strategies that people leveraged to maintain recovery during the first year of the COVID-19 pandemic. Mindful of the potential for variation by gender observed in some COVID-19 studies, we assessed differences between women and men. As an exploratory study there were no a priori hypotheses.

## Materials and methods

### Study design and sample

We used a convergent mixed-methods approach [37]. Specifically, quantitative and qualitative data were collected via the same cross-sectional survey questionnaire and were analyzed independently but at the same time. To synthesize findings, we compared and contrasted results, seeking commonalities and differences across the quantitative and qualitative results. Neither type of data was given priority over the other. We chose the mixed-methods design as we believed it would provide better depth of knowledge than either approach alone, particularly as the COVID pandemic was a novel experience.

Participants were recruited from KnowledgePanel, a probability-based cohort of non-institutionalized adults (age 18 and older) maintained by Ipsos for ongoing internet-based research. Details of KnowledgePanel are available elsewhere (https://www.ipsos.com/en-us/solutions/public-affairs/knowledgepanel). In fall 2020, Ipsos drew a general population sample of KnowledgePanel members and an oversample of racial and ethnic minority panel members for our survey about recovery from alcohol problems. The survey was available in English or Spanish. To be eligible to participate, panel members had to be age 18 or older and self-identify as a person in recovery or with a resolved alcohol problem (i.e., affirming that they have "taken care of, gotten over, or resolved a previous drinking problem"). Abstinence was not required; however, the study's overarching aim was to recruit a sample of adults in stable recovery. Therefore, current drinkers were screened for hazardous drinking using the three-item AUDIT-C [38, 39]. Potential participants with scores indicative of hazardous drinking ($\geq$4 for men; $\geq$3 for women) were excluded as it was assumed to be incongruent with stable recovery. In total, 31,386 KnowledgePanel members were invited to participate, of whom 17,622 completed an eligibility screening (56% response rate). Of those respondents, 1,637 met eligibility criteria and provided informed consent to participate. As the questionnaire was online, participants read an informed consent document and clicked to affirm agreement before proceeding to the questionnaire. During data preparation, we discovered that 145 respondents failed to meet all eligibility criteria (e.g., reported zero lifetime AUD symptoms or did not self-identify as in recovery or having had a prior alcohol problem). Subsequently we excluded

them, yielding an analytic sample of 1,492 adults. As compensation, survey participants received 20,000 KnowledgePanel points, worth approximately $20. Study materials and procedures were reviewed and received ethical approval by the University of Iowa Institutional Review Board (#201905208).

## Measures

To assess strategies to maintain recovery, participants were asked to endorse items on a choose-all-that-apply list of possible coping responses, producing the quantitative data for this analysis. We adapted the list of coping responses from the Environmental Influences on Child Health Outcomes survey, available at the Disaster Research Response Resources Portal (https://tools.niehs.nih.gov/dr2/). We chose these items in order to align our survey with an established measure of coping behaviors to increase comparability of our results. In addition, a single open-ended question generated qualitative data for this analysis. Participants were asked: "What has been most helpful in sustaining your recovery during the coronavirus/ COVID-19 outbreak?" Of the full sample, 1,236 (83%) participants provided an open-ended response; however, we excluded 228 participants because their responses did not address the question prompt, were ambiguous, or expressed no concern about the impact of the COVID-19 pandemic on recovery, resulting in text responses from 1,008 participants retained for the qualitative analysis.

Lifetime AUD symptoms were assessed via 11 items drawn from the National Epidemiological Survey on Alcohol and Related Conditions, which conformed to DSM-5 diagnostic criteria (https://www.niaaa.nih.gov/research/nesarc-iii/questionnaire). Affirmative responses were summed to create a count of symptoms (range 1–11). In accordance with DSM-5 criteria, we created a four-level classification of lifetime AUD severity: sub-clinical (1 symptom); mild (2–3 symptoms); moderate (4–5 symptoms); and severe (6 or more symptoms). Following National Institutes of Health classifications [40], length in recovery was reported by participants as a categorical variable: early recovery (<1 year); intermediate recovery (1–5 years); or long-term recovery (>5 years). We further classified participants into three recovery groups based on self-reported lifetime use of 14 different services. The groups consisted of treated recovery (any use of specialty services, such as in-patient or out-patient rehabilitation), assisted recovery (any use of lay services, such as mutual-help groups, and no use of specialty services), and independent recovery (no use of specialty nor lay services). Finally, we used nine sociodemographic variables to describe the sample: age; gender; race/ethnicity; educational attainment; employment status; relationship status; household poverty status; the presence of minor children in the household; self-rated health; and quality of life.

## Analysis

First, we summarized demographic characteristics and the checklist of coping responses using unweighted frequency counts and percentages. Ipsos assigns all KnowledgePanel members geodemographic weights (i.e., controlling for age, gender, education, income, Census region, and metropolitan status within racial/ethnic groups) so that results may be representative of the US adult population following benchmark distributions from the March 2020 Current Population Survey and the 2018 American Community Survey. Given our interest in variation by gender, we tested differences between women and men using survey weighted chi-square and ANOVA tests. All quantitative analyses were performed with the statistical software R using the 'survey' package, version 4.0 [41, 42], with a critical alpha of 0.05. Second, we analyzed the qualitative data through a thematic analysis of responses using Dedoose software [43]. Following established procedures [44], we developed inductive codes and used them to

label the data, identified themes and grouped them into related clusters, elaborated the meaning of each cluster, and investigated relationships between the codes and clusters. We contrasted the application of codes among women and men to explore differences by gender. The qualitative analysis was an iterative process that involved multiple rounds of reading, coding, and interpreting data. Memos and notes ensured rigor and provided an audit trail to review analytic decisions. Questions and disagreements were resolved through discussion among authors until consensus was reached. None of the authors identifies as a person in recovery; throughout the analysis, we were mindful of how the lack of direct experience may have affected our interpretation of results.

## Results

### Sample characteristics

Table 1 shows study participants' demographic and alcohol problem characteristics. Overall, majorities of the sample met criteria for severe lifetime AUD (72.9%), reported being in recovery more than five years (75.5%), and were classified in the independent recovery group (i.e., having used neither specialty services nor mutual-help groups in their lifetime; 59.7%). In addition, majorities were male (69.0%), White (65.0%), married or cohabitating (59.7%), and employed full- or part-time (54.0%). Close to half of the sample was age 60 years or older (44.9%). Smaller proportions of participants reported excellent or very good health (36.7%), had a college degree (30.1%), and reported household incomes less than the federal poverty limit (17.6%) or near (i.e., 100%-200%) the federal poverty limit (19.5%). We did not detect any significant gender differences in the distributions of educational attainment, self-rated health, recovery length, and recovery group; however, women and men differed significantly on all other demographic variables. Notably, greater proportions of men than women were employed full- or part-time and were married or cohabitating. Conversely, greater proportions of women than men had household incomes below the federal poverty limit, had minor children in the household, and reported very good or pretty good quality of life.

### Quantitative checklist of coping responses

Table 2 shows endorsements of items in the checklist of COVID-related coping strategies for the full sample and by gender, using weighted percentages to make generalizable inferences. The rank ordering of coping strategies was quite similar for women and men with a few notable differences. For example, nearly half of women and approximately one-third of men reported talking with friends and family by phone, text, or video (49.7% vs. 36.1%, $p < .001$). Although endorsed at lower levels, a much greater proportion of women than men endorsed spending increased time reading books or doing activities like puzzles and crosswords (29.5% women vs. 16.8% men, $p < .001$). Women were also significantly more likely than men to report using tobacco (8.3% women vs. 5.0% men, $p = .03$) and doing "something else" (7.9% women vs. 4.8% men, $p = .04$). In contrast, a much greater proportion of men than women reported not doing anything to cope with the COVID-19 pandemic (32.0% men vs 17.1% women, $p < .001$). Although alcohol use was very low overall, men were slightly more likely than women to report drinking as a coping strategy (2.2% men vs. 0.7% women, $p = .03$).

### Qualitative responses about coping

Our inductive, thematic analysis of text responses to the open-ended question about sustaining recovery during the COVID-19 pandemic yielded 26 distinct codes, which we grouped into 10 thematic clusters. Fig 1 shows the clusters, with sizes proportional to the number of statements

**Table 1. Demographic characteristics of a US national sample of adults with a resolved alcohol problem.**

| | Full sample (n = 1492) | Women (n = 463) | Men (n = 1029) | p |
|---|---|---|---|---|
| Age | | | | |
| 18–29 years | 89 (6.0%) | 40 (8.7%) | 49 (4.8%) | 0.01 |
| 30–44 years | 294 (19.7%) | 99 (21.4%) | 195 (19.0%) | |
| 45–59 years | 439 (29.4%) | 134 (28.9%) | 305 (29.6%) | |
| ≥60 years | 670 (44.9%) | 190 (41.0%) | 480 (46.6%) | |
| Race/ethnicity | | | | |
| White, non-Hispanic | 970 (65.0%) | 311 (67.2%) | 659 (64.0%) | |
| Black, non-Hispanic | 165 (11.1%) | 61 (13.2%) | 104 (10.1%) | 0.01 |
| Hispanic, any race | 245 (16.4%) | 56 (12.1%) | 189 (18.4%) | |
| Multiple or other races | 112 (7.5%) | 35 (7.6%) | 77 (7.5%) | |
| Educational attainment | | | | |
| Less than high school | 110 (7.4%) | 40 (8.6%) | 70 (6.8%) | 0.07 |
| High school diploma | 373 (25.0%) | 129 (27.9%) | 244 (23.7%) | |
| Some college | 560 (37.5%) | 173 (37.4%) | 387 (37.6%) | |
| Bachelor's degree or higher | 449 (30.1%) | 121 (26.1%) | 328 (31.9%) | |
| Employment status | | | | |
| Employed full- or part-time | 805 (54.0%) | 218 (47.1%) | 587 (57.0%) | <0.001 |
| Unemployed | 67 (4.5%) | 28 (6.0%) | 39 (3.8%) | |
| Out of the labor force | 620 (41.6%) | 217 (46.9%) | 403 (39.2%) | |
| Relationship status | | | | |
| Married or cohabitating | 890 (59.7%) | 244 (52.7%) | 646 (62.8%) | <0.001 |
| Widowed, divorced or separated | 346 (23.2%) | 141 (30.5%) | 205 (19.9%) | |
| Never married | 256 (17.2%) | 78 (16.8%) | 178 (17.3%) | |
| Household poverty status | | | | |
| <100% federal poverty level | 262 (17.6%) | 114 (24.6%) | 148 (14.4%) | <0.001 |
| 100%-200% federal poverty level | 291 (19.5%) | 98 (21.2%) | 193 (18.8%) | |
| >200% federal poverty level | 939 (62.9%) | 251 (54.2%) | 688 (66.9%) | |
| Minor children in household | | | | |
| No | 1163 77.9%) | 341 (73.7%) | 822 (79.9%) | <0.01 |
| Yes | 329 (22.1%) | 122 (26.3%) | 207 (20.1%) | |
| Self-rated health | | | | |
| Excellent or very good | 465 (36.7%) | 127 (33.0%) | 338 (38.3%) | 0.16 |
| Good | 488 (38.5%) | 153 (39.7%) | 335 (38.0%) | |
| Fair or poor | 314 (24.8%) | 105 (27.3%) | 209 (23.7%) | |
| Quality of life | | | | |
| Very good or pretty good | 1124 (75.6%) | 319 (69.4%) | 805 (78.5%) | <0.001 |
| Good and bad about equal | 294 (19.8%) | 115 (25.0%) | 179 (17.5%) | |
| Pretty bad or very bad | 68 (4.6%) | 26 (5.7%) | 42 (4.1%) | |
| Lifetime AUD severity | | | | |
| Sub-clinical (1 symptom) | 52 (3.5%) | 15 (3.2%) | 37 (3.6%) | 0.04 |
| Mild (2–3 symptoms) | 146 (9.8%) | 60 (13.0%) | 86 (8.4%) | |
| Moderate (4–5 symptoms) | 207 (13.9%) | 58 (12.5%) | 149 (14.5%) | |
| Severe (6 or more symptoms) | 1087 72.9%) | 330 (71.3%) | 757 (73.6%) | |
| Recovery length | | | | |
| <1 year | 66 (4.4%) | 19 (4.1%) | 47 (4.6%) | 0.38 |
| 1–5 years | 289 (19.4%) | 99 (21.4%) | 190 (18.5%) | |
| >5 years | 1127 (75.5%) | 340 (73.4%) | 787 (76.5%) | |

*(Continued)*

**Table 1.** (Continued)

|  | Full sample (n = 1492) | Women (n = 463) | Men (n = 1029) | p |
|---|---|---|---|---|
| Recovery group |  |  |  |  |
| Independent | 891 (59.7%) | 295 (63.7%) | 596 (57.92%) | 0.12 |
| Assisted | 226 (15.1%) | 61 (13.2%) | 165 (16.0%) |  |
| Treated | 375 (25.1%) | 107 (23.1%) | 268 (26.1%) |  |

Note: Both frequency counts and percentages are unweighted.

associated with each one, and Fig 2 shows the individual codes by frequency. In the text below, we summarize the thematic clusters and describe their most important constituent codes. Further details are provided in S1 Appendix, which presents all code definitions by theme with illustrative quotations.

**Theme: Staying connected.** This was by far the most frequently invoked theme, and statements about family accounted for the majority of coded excerpts (73%). Participants regularly referenced talking to, staying in touch with, or spending time with family. Comments often highlighted family in a general way, for example "maintaining communication with family and friends," or "spending more time with my family and playing with them." However, there were also mentions of specific family members, such as spouses, children, or grandchildren. One participant referenced "my wonderful husband, who listens and helps with my mental health," while another brought up "playing a daily game with my daughter." Notably, women's responses had a much more pronounced emphasis on children than men's responses. "My kids keep me busy," said one female participant, while another woman endorsed "being there for my children" as a factor that helped her maintain recovery. Support and encouragement

**Table 2. Endorsements of coping strategies during the COVID-19 pandemic in a US national sample of adults with a resolved alcohol problem.**

|  | Full sample (n = 1482) | Women (n = 463) | Men (n = 1029) | p |
|---|---|---|---|---|
| Increased television watching or other screen time activities (video games, social media) | 659 (42.7%) | 235 (46.8%) | 424 (40.6%) | 0.06 |
| Talking with friends and family by phone, text, or video | 610 (40.8%) | 245 (49.7%) | 365 (36.1%) | <0.001 |
| Exercising | 471 (31.2%) | 158 (34.6%) | 313 (29.4%) | 0.09 |
| Spending time in nature | 416 (27.9%) | 154 (30.8%) | 262 (26.3%) | 0.13 |
| Eating more often, including snacking | 336 (22.9%) | 123 (25.1%) | 213 (21.8%) | 0.23 |
| Increased time reading books or activities like puzzles and crosswords | 356 (21.2%) | 163 (29.5%) | 193 (16.8%) | <0.001 |
| Engaging in more family activities, games, or sports | 235 (18.0%) | 80 (20.4%) | 155 (16.8%) | 0.19 |
| Meditation or mindfulness practices | 270 (17.0%) | 103 (19.5%) | 167 (15.7%) | 0.12 |
| Engaging in creative pursuits, like writing, creating art, or making music | 190 (12.8%) | 81 (15.4%) | 109 (11.5%) | 0.10 |
| Using cannabis (smoking, vaping, eating) or cannabidiol (CBD) | 108 (7.1%) | 42 (8.6%) | 66 (6.3%) | 0.17 |
| Doing volunteer work or helping others | 118 (6.6%) | 43 (8.3%) | 75 (5.7%) | 0.09 |
| Using tobacco (smoking or vaping) | 94 (6.2%) | 38 (8.3%) | 56 (5.0%) | 0.03 |
| Talking to my healthcare providers more frequently, including mental health providers like therapist, psychologist, or counselor | 94 (6.1%) | 49 (8.0%) | 45 (5.2%) | 0.08 |
| Something else | 92 (5.9%) | 32 (7.9%) | 60 (4.8%) | 0.04 |
| Drinking alcohol | 25 (1.7%) | 4 (0.7%) | 21 (2.2%) | 0.03 |
| I have not done anything to cope with the coronavirus/COVID-19 outbreak | 396 (26.9%) | 74 (17.1%) | 322 (32.0%) | <0.001 |

Note: frequency counts are unweighted; percentages are weighted.

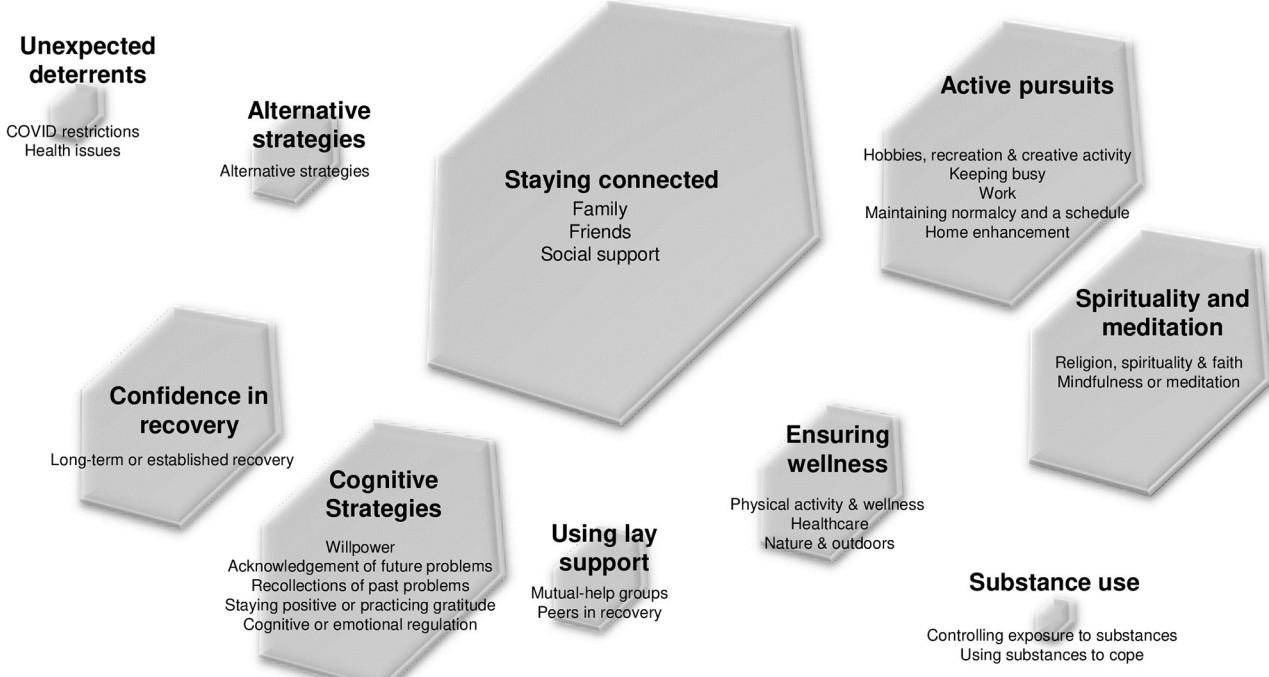

**Fig 1. Thematic clusters.** Note: The size of the polygon reflects the relative frequency of excerpts included in the theme.

received from and given to family was also highlighted by participants, with one participant stating "[my family] sticks together and helps each other." Another participant referred to "a pact I made with my brother that both of us would no longer use alcohol," as his motivation for sustaining recovery during COVID. Finally, taking on responsibilities for their family's wellbeing was noted by several participants, as highlighted by one man focusing on "helping my wife who has cancer" during the pandemic.

**Theme: Active pursuits.** This was the second most frequent theme, comprised of five constituent codes that were similarly sized and which appeared to reflect complementary dimensions of the central principle of keeping oneself occupied to discourage drinking. For example, statements such as, "fixing up the things around the house, such as painting, yard-work, and fixing the fence" was one man's comment about what helped him maintain recovery during the pandemic. Other participants mentioned engaging in hobbies such as "reading," "being creative by writing," and "being able to travel in my RV" as successful ways to maintain their recovery during the pandemic. Participants also referenced preoccupation with work as a means of minimizing opportunities to use alcohol with responses like, "Staying in my normal routine. Note I work a full-time job and [am] self-employed part-time" and "working 16 hours a day keeps me sober." Furthermore, participants also felt that "conducting life as normal" and "keeping to a routine" were important to their recovery. Lastly, even the nondescript actions of "staying busy" and "just keeping my mind occupied" were important elements of participants' strategies to maintain recovery during the pandemic.

**Theme: Cognitive strategies.** This theme reflected attempts to regulate thoughts and emotions, with mentions of willpower accounting for the largest share of excerpts (46%) among the five constituent codes in this thematic cluster. This often appeared as expressions of determination, commitment, or confidence in one's ability to maintain their recovery as the

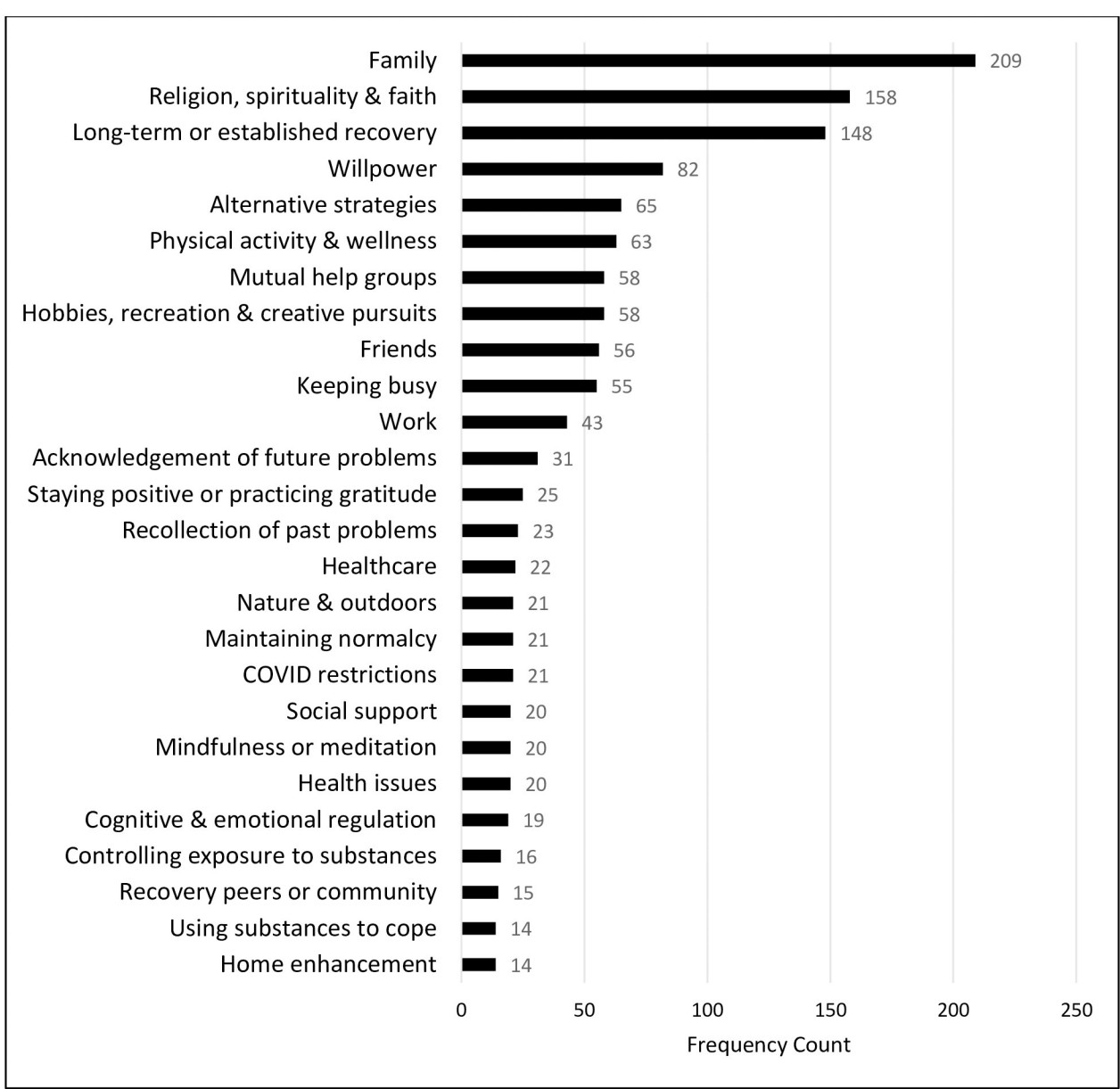

**Fig 2. Code frequencies.**

most helpful factor during the pandemic. This theme also included actions such as "just saying no," as well as responses in which participants highlighted "me/myself/the person in the mirror." Both men and women referenced their own strength, a firm mindset, and indications that they had made a personal choice that they would uphold. "I own my decision to not drink" was one woman's comment. However, men brought up willpower in their responses much more frequently than did women, and some comments suggested beliefs in their individual capability for—and control of—successful recovery as personal attributes. "Belief in my ability to overcome any difficulty put in my path," one man declared, "improvise, adapt, overcome!" Another noted that "nothing makes me drink anymore. I have complete self-control."

**Theme: Spirituality & meditation.** This theme was driven to a very large extent by a code for "religion, spirituality, and faith," which captured 89% of all excerpts included here. Comments emphasized religious institutions, faith practices, and personal spirituality. Some participants underlined their reliance on and trust in God for support and safety: "A desire to not drink and praying about it," stated one participant, while another explained that he has "trust in the Lord to watch over me and those I love!" Praying, attending church, or reading the Bible were also often referenced as beneficial activities in sustaining recovery during the pandemic. Interestingly, men's references to faith were more varied and nuanced than women's references. In addition, men's comments suggested that faith was something they needed to actively maintain: "It's like jogging every day to stay in shape. Pray and read the word to stay in shape!" In contrast, women tended to emphasize God as a distinctive and strong source of support. Among statements, one woman wrote "I have the Lord in my life and that's all need for support" while another wrote "He [Jesus] will never leave me or forsake me. He loves me and you."

**Theme: Confidence in recovery.** Expressions of confidence in recovery constituted the fifth most frequent thematic cluster, which was driven by statements about "long-term and established recovery"—the third most frequently applied code. At times, these statements were coupled with proclamations that participants no longer experienced the need, desire, or temptation to drink as a result of the years they had spent being abstinent or problem-free. As one participant explained, "I've been alcohol-free so long I don't have any desire to drink. . .COVID is not a factor." Some participants also noted that their long recovery gave them confidence in their self-control. Alternatively, if they did not pursue abstinence, their long recovery gave them a strong awareness of the drinking limits they wished to maintain. One participant stated that "I have had control over alcohol use for 18 years; I can have one drink and stop and go a year before I have another drink." Some suggested that in view of their long sobriety, marked by occasional light drinking in some cases, they no longer considered themselves to be in recovery: "I am not recovering. I have a beer on occasion. My alcohol problem started and ended 40-plus years ago with total abstinence for 20-plus years and now with an occasional beer."

**Theme: Ensuring wellness.** This theme reflected a focus on physical health. Indeed, exercise and physical activity accounted for the majority of statements under this theme (59%). This was exemplified in statements such as "exercise–this has become an addiction, albeit a good one" and "getting out of the house and doing something physical, like walking, hiking, [or] swimming." Participants often noted the natural setting in which they engaged in physical activity being important to them as well with one participant stating, "enjoying nature in my backyard" and another mentioning, "exercise in nature." In line with maintaining physical health, participants also detailed engagement with their health care providers in statements like, "Being aware of my health and seeing my providers in a timely manner" as well as "Support from my health team." Under this theme, men were more likely to provide responses about nature and physical activity, while women were more likely to frame their responses around interactions with their health care providers.

**Theme: Alternative strategies.** Although this emerged as an important theme, it was difficult to summarize. The common element was use of highly individualized strategies that were not captured by any other codes. Some comments appeared to reflect characteristics of the COVID pandemic, such as one man's statement that he "focuses on things that are needed, such as finding foods and supplies." Despite the diversity of responses, there were multiple comments about limiting exposure to news, such as "keeping exposure to mainstream media to an absolute minimum," or mentions of pursuing educational opportunities, such as "start studying at the university." Other repeated comments focused on relaxation, such as "just

relax and don't have any worries," and embracing alone time, for example "appreciating the time away from people and negativity." We noted a gender imbalance, with two-thirds of alternative strategies excerpts coming from men; however, there was no discernable pattern of men's versus women's responses.

**Theme: Using lay support.** A relatively minor theme, use of lay support for recovery was dominated by comments about mutual-help groups and peers in recovery. Specifically, participants talked about staying engaged in 12-step groups through remote meetings. In addition, one participant expressed the benefit of remote meetings: "My AA home group has a daily Zoom meeting. As a community, I feel we have gotten so much closer. It's such a relief from any COVID-related anxiety." Other statements referred to individual activities, such as "reading the Big Book," or alternative ways to maintain connections to peers, such as "messaging friends in the program."

**Theme: Unexpected deterrents.** This minor theme focused on COVID-related restrictions that limited opportunities to drink and health issues that deterred drinking. Participants found that bar closures and social distancing requirements were unexpectedly beneficial. One participant explained that "inaccessibility to social drinking has curbed lots of relapses," and another stated that "COVID-19 has actually helped keep me from dealing with social situations where I would be more likely to want to drink." In addition, health issues, such recovery from surgery, diabetes, alcohol's interactions with prescription medications, and fear of developing future health issues were given as reasons participants avoided alcohol. We detected few gender differences; however, women stated that health issues prompted them to not only quit drinking but also avoid over-drinking, whereas men only spoke about quitting alcohol entirely due to health issues.

**Theme: Substance use.** This theme received the fewest statements, yet there were a sufficient number to identify it as a strategy. Most responses centered around limiting physical proximity to alcohol, such as "Not having it in my home" or "The same thing that enabled me to quit years ago. I quit being a bartender and do not go to bars for any reason whatsoever." Participants who still drank alcohol set situational or drink limits, such as "Continuing to limit my drinking to occasional activities with friends once a month or less and no more than one or two drinks." A minority of responses suggested cannabis use for coping, often with very brief responses such as "bud," "weed," or "access to cannabis." One participant elaborated by stating "Smoking pot and defying [governor's] COVID lockdown orders."

## Comparing quantitative and qualitative results

We compared results from the quantitative checklist and qualitative text responses to gauge similarities and differences in strategies. Among similarities, family was the most frequent qualitative code, and communication with family and friends and engaging in family activities were among the top most frequently endorsed quantitative items. Active pursuits was the second most important qualitative theme, while elements of that theme (e.g., reading books, creative pursuits) were the sixth and ninth most frequently endorsed quantitative coping strategies. Substance use was a relatively infrequent coping strategy, but it did appear in both qualitative and quantitative results. Similarly, minorities of participants reported not being concerned about the COVID pandemic (10% of open-ended responses) and not doing anything in particular to cope (27% endorsement in the checklist). In terms of differences, religion, spirituality, and faith was the second most frequent code in the qualitative analysis, however, it did not appear in the quantitative checklist as an option. In addition, a number of qualitative codes emerged that were not included in the quantitative checklist (e.g., willpower, mutual-help groups, acknowledgement of future problems). This was likely because the

checklist was developed by others for general population use during the COVID-19 pandemic, not specifically about strategies to maintain recovery among people with a previous alcohol problem.

## Discussion

There has been growing concern about deteriorating mental health and corresponding increases in alcohol consumption during the COVID-19 pandemic. At the same time, there has been scant attention to the pandemic's impact on people in recovery from AUD. In response, this study sought to understand how the COVID-19 pandemic affected recovery from alcohol use disorder. Specifically, we used a mixed-methods approach to identify strategies to maintain recovery in a national sample of US adults with a resolved alcohol problem. Notably, more than half of our sample consisted of people who had never obtained specialty treatment or used mutual-help groups, a hard-to-reach and understudied population. In addition, we paid particular attention to differences between women and men given recent reports that women's drinking patterns changed in ways different than men's during the COVID-19 pandemic, including sharper increases in heavy drinking [20, 45]. While we found only a very small gender difference in the very low prevalence of drinking to cope during the pandemic, we did find some gender differences in the types of strategies reported (e.g., women's greater tendency to talk with family and friends, and men's greater tendency to report not doing anything to cope with COVID-19 or to emphasize willpower and self-reliance).

Our main finding is that staying connected to others was a key strategy for recovery maintenance during the first year of the pandemic, and that a variety of social relationships and roles were relevant. Family was particularly important, both in terms of providing support to participants but also as the recipients of participants' care and help to others. Regarding family, we noted a gender difference in which women emphasized the importance of children more than men. Friends and peers in recovery played a role too. Importantly, social connections were not limited to close, interpersonal relationships. Staying connected as a strategy could also take the form of participating in organizations, such as mutual-help groups and faith communities. These findings are highly congruent with previously published research. Notably, a recent systematic review found that supportive social ties were consistently associated with reduced relapse risk [46], and other research has found that helping others reduces risk for binge drinking among those who continue to drink following AUD treatment [47]. Previous research also suggests some intriguing variations in the role and nature of social ties. For example, Timko and colleagues found that patients receiving treatment for AUD who did not have a concerned family member or other person did not necessarily have poorer likelihood of recovery [48]. Also, in a qualitative study of adults recently discharged from treatment, participants preferred support through Alcoholics Anonymous over family members because peers in recovery were perceived as a better source of social support [49]. Another study of persons living in a recovery community contrasted support from family members versus friends, finding an inverse association between general support from friends and subsequent alcohol use but no such association between family support and drinking outcomes [50]. Thus, while family members may be important to recovery for many people, other social ties and relationships can be instrumental in facilitating recovery through both the receiving and giving of support. There may be value in looking beyond family members as sources of recovery support, particularly during the early recovery period when family relationships may be strained for some individuals. Future research should assess both the relative importance, timing, and types of recovery support through different social ties.

A striking finding is that many of the strategies we identified are consistent with established therapeutic modalities. For example, under the qualitative theme of cognitive strategies, activities such as recalling past problems and anticipating future problems if one resumes drinking are quite similar to Contingency Management principles, such as negative reinforcement and negative punishment [51]. Some activities under the theme of active pursuits may be comparable to Cognitive Behavioral Therapy's redirection towards pleasant activities, in which engaging in activities a person enjoys or is passionate about reduces the temptation to drink and reinforces the ability to enjoy life without alcohol [52]. Furthermore, both meditation or mindfulness practices as well as maintaining normalcy and a schedule reflect tenets of Acceptance and Commitment Therapy, such as being present, aligning actions with values, and focusing pragmatically on what one is able to control [53]. Considered together, we find the congruence between these strategies reported by our participants—the majority of whom had never utilized specialty treatment or mutual-help groups—and components of established psychotherapies to be particularly noteworthy. It suggests that some elements of psychotherapies may be well received by treatment-naïve individuals as these mirror recovery-oriented behaviors that emerge organically. Moreover, it hints at the potential value of novel intervention strategies, such as developing lay guides for persons who prefer not to seek services or leveraging existing social connections to deliver components of existing psychotherapies to support recovery.

Unexpectedly, willpower emerged as an important aspect of the cognitive strategies theme, accounting for nearly half of the theme's excerpts. This parallels other qualitative studies that have reported a prominent role for willpower in AUD recovery [54, 55]. Seeking to understand it better, we wondered if our results may have been driven by the sample composition. We conducted a posthoc analysis to assess whether willpower was more often invoked by participants in the independent recovery group—the majority of our sample—compared to those in the assisted or treated recovery groups. There was no difference in mentions of willpower across the three recovery groups, reported by 4%-6% of all groups (p = 0.19). As it was unrelated to group membership, it may have reflected a personality trait. Some people pursuing recovery may turn inward (i.e., drawing upon individual strengths and skills) while others may turn outward (i.e., seeking support from family, friends, and peers in recovery). Interestingly, strong statements about personal willpower were more often voiced by men than women, which may reflect gender differences in help-seeking attitudes and tendencies. Assessing a person's recovery orientation, which might be a reflection of their health locus of control [56], sense of self-efficacy [57], or willingness to seek external support, may enable better tailoring of supportive services. However, we caution that conceptualizing alcohol addiction and relapse as a failure of personal willpower carries high risk of perpetuating stigma [58]. Despite its emphasis by some of our participants, willpower is unlikely to serve as an appropriate intervention target; but to the extent that it overlaps with self-efficacy (which is a component of recovery capital and is indeed modifiable) novel interventions to improve recovery self-efficacy may align with some people's emphasis of willpower as a strategy for recovery.

Among minority findings, substance use was rare but nevertheless present. For example, alcohol use was endorsed by approximately two percent of respondents who completed the quantitative coping checklist, and using substances to cope was a qualitative code applied to just 14 statements. This may reflect our eligibility criteria; people with indications of hazardous alcohol use were not eligible to participate in the study.

One counterintuitive finding stands out. A minority of participants (n = 21, 2% of open-ended responses) reported that COVID-19 restrictions, such as stay-at-home orders that limited social interactions, were helpful to maintain their recovery. This is in striking contrast to

our main finding that staying connected was the most frequently used strategy. The unexpected benefit may be due to the separation from people and places that could trigger drinking. Indeed, a recent international review found that both physical environment and social context have been associated with excessive drinking [59], and this relationship could extend to relapse risk. This unexpected benefit of pandemic restrictions may be consistent with treatment approaches that emphasize avoiding certain people and places that jeopardize recovery. However, as with willpower, this finding presents a challenge for research and practice. On the face of it, social separation and physical distancing would make poor intervention activities, particularly without bolstering social support in other ways. In the short-term, what may be needed is an intervention to develop the skills to differentiate risky social contexts from supportive social ties. In the long-term, it may be necessary to develop plans to ensure the availability of social support for recovery through alternative channels during natural disasters or other large-scale catastrophes.

In terms of other practice implications, several of our participants' strategies align with recommendations for providing services to substance-using populations during the COVID-19 pandemic that were identified in a recent review [60]. For example, the common strategy of staying connected to maintain one's recovery is reflected in recommendations to "create a buddy and self-support system with someone trusted and to reach out for extra help if needed." In addition, several other qualitative themes emerging from this study are reflected in recent recommendations for services to substance-using populations (e.g., maintaining normalcy and a routine; drawing upon general social support and peers in recovery; using mutual-help groups). While it is heartening to see that many recommendations align with the actual practices of people in recovery that we identified, some of the important themes that we found (e.g., religion and spirituality as helping recovery) have not been included in guidelines. Thus, our findings may inform future efforts to support recovery by extending the repertoire of potential strategies and partners (e.g., faith communities that provide mental health first aid services during a disaster).

While this study has several notable strengths, including the use of a national sample, timely data collection in the first year of the COVID-19 pandemic, and an open-ended query that allowed participants to voice their strategies without use of a priori response categories, our findings should be considered in light of several caveats and limitations. First, the qualitative data may not have captured the experiences of all people in recovery due to differential responses. Participants were free to skip the open-ended question or may have had different motivations to respond to our question. Indeed, 17% participants did not provide any answer to the prompt, and of those who did provide responses, 18% were excluded as irrelevant or ambiguous. Second, participant characteristics may have influenced our findings. Notably, the majority of our sample reported being in long-term recovery, and these participants might benefit from higher levels of protective factors, such as well-developed coping strategies, that have sustained their recovery. Thus, it is possible that persons who are newly attempting recovery would adopt different strategies, and these strategies may not have been identified in our study. Indeed, a study with a different sample composition may have found a different pattern of strategies for recovery. Third, as noted above, the study's eligibility criteria excluded potential participants with indications of hazardous drinking, and thus the sample represents individuals who have some success in managing their drinking. This may have artificially decreased reports of drinking to cope, which were reported at very low levels, or prevented us from capturing the recovery strategies of people in a relapse episode. Fourth, our findings are bound by time and place. Specifically, this study focused on recovery during the first year of the COVID-19 pandemic in the US. Strategies to maintain recovery during other large-scale stressors may differ in other periods and locations.

## Conclusions

Our study identified several key strategies that adults used to maintain recovery from alcohol problems during the COVID-19 pandemic. These findings may inform future research on the mechanisms of successful recovery as well as identify potential intervention activities for recovery support during other catastrophic events (e.g., ensuring the communications infrastructure to facilitate exchanges of social support; training lay partners to provide mental health first aid).

## Supporting information

**S1 Appendix. Code definitions by theme.**
(DOCX)

## Acknowledgments

We thank Grant Brown, PhD, for maintaining the quantitative data set and producing the summaries shown in Tables 1 and 2.

## Author Contributions

**Conceptualization:** Paul A. Gilbert.

**Data curation:** Loulwa Soweid.

**Formal analysis:** Paul A. Gilbert, Loulwa Soweid, Paul J. Holdefer, Sarah Kersten, Nina Mulia.

**Project administration:** Loulwa Soweid.

**Supervision:** Paul A. Gilbert.

**Writing – original draft:** Paul A. Gilbert, Loulwa Soweid, Paul J. Holdefer.

**Writing – review & editing:** Sarah Kersten, Nina Mulia.

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
