## [Decision Letter · Decision Letter 0]

10 Feb 2023

PONE-D-22-28505Strategies to maintain recovery from alcohol problems during the COVID-19 pandemic: Insights from a mixed-methods national survey of adults in the United StatesPLOS ONE

Dear Dr. Gilbert,

Thank you for submitting your manuscript to PLOS ONE. After careful consideration, we feel that it has merit but does not fully meet PLOS ONE’s publication criteria as it currently stands. Therefore, we invite you to submit a revised version of the manuscript that addresses the points raised during the review process.

We look forward to receiving your revised manuscript.

Kind regards,

Asrat Genet Amnie, MD, EdD, MPH, MBA

Academic Editor

PLOS ONE

Journal Requirements:

This study was supported by the National Institutes of Health (R01AA027266). The content is solely the responsibility of the authors and does not necessarily represent the official views of the National Institutes of Health.

Additional Editor Comments:

Please provide point-by-point response to reviewer’s comments and concerns. Please include evidence of IRB approval and/or a statement of ethical clearance. Also please clarify your sample population and the source population from which the sample was drawn. The number of themes should be consistent in the narratives and tables throughout the manuscript.

Reviewers' comments:

Reviewer's Responses to Questions

**Comments to the Author**

1. Is the manuscript technically sound, and do the data support the conclusions?

Reviewer #1: Yes

Reviewer #2: Yes

2. Has the statistical analysis been performed appropriately and rigorously? 

Reviewer #1: Yes

Reviewer #2: Yes

3. Have the authors made all data underlying the findings in their manuscript fully available?

Reviewer #1: Yes

Reviewer #2: Yes

4. Is the manuscript presented in an intelligible fashion and written in standard English?

Reviewer #1: Yes

Reviewer #2: Yes

5. Review Comments to the Author

Reviewer #1: Thank you for an excellent and clear presentation of this wonderfully readable piece of work. My only main feedback would be if you would rework the abstract ever so slightly to more clearly indicate where the sample was drawn from. It becomes very clear in the text but the abstract does leave one wondering. The other feedback I might add is if you could say very very briefly why you went with the survey tools you chose. It need not be long, something to the effect of "it has been used frequently in this population or similar populations and is reliable etc etc". I would argue that the findings here are actually not applicable to just dramatic events but to recovery literature more broadly. I really enjoyed reading this, have learned a few new tings and reinforced previously understood areas as well. Congrats!

Reviewer #2: Thank you for the opportunity to review this very interesting and relevant paper.

This is very well written paper that will be of interest to the readership of the journal.

My comments/suggestions are minor and are related to some inconsistencies or need for clarification.

Abstract.

Line 36

You mentioned 10 themes but didn't list all 10 in the abstract.

Also, in the main text there are only 7 themes and in the figure 2, 9 themes. Please make sure you correct these sections so there is consistency. 10 themes is a lot and seems that some of the 9 themes in figure 2 a very small and perhaps there is some overlap so they can be merged?

Methods

line 148, i presume this is ethical approval? if so, please be explicit.

line 163. If lifetime Aud symptoms were collected before the survey, should come before describing the survey.

line 166. was this classification arbitrary or based on previous scales?

line 188. better to say in statistical package/software R, otherwise confusing for those that are not familiar with it. also for consistency with saying dedoose software.

193. Did you use thematic analysis? Braun and Clarke?

Results:

Table 1: bracket missing at very good or good QoLand recovery length over 5 years.

Themes:

in main text 7 themes, in figure 2, 9 themes. please be consistent and delete remove/merge as necessary.

6. PLOS authors have the option to publish the peer review history of their article (what does this mean?). If published, this will include your full peer review and any attached files.

Reviewer #1: No

Reviewer #2: **Yes: **Dr Mariyana Schoultz

---

## [Author Response · Author response to Decision Letter 0]

10 Mar 2023

All changes have been detailed in the Response to Reviewers document.

---

## [Editor Report · Decision Letter 1]

3 Apr 2023

Strategies to maintain recovery from alcohol problems during the COVID-19 pandemic: Insights from a mixed-methods national survey of adults in the United States

PONE-D-22-28505R1

Dear Dr. Author(s), 

We’re pleased to inform you that your manuscript has been judged scientifically suitable for publication and will be formally accepted for publication once it meets all outstanding technical requirements.

Kind regards,

Asrat Genet Amnie, MD, EdD, MPH, MBA

Academic Editor

PLOS ONE

---

## [Editor Report · Acceptance letter]

10 Apr 2023

PONE-D-22-28505R1 

Strategies to maintain recovery from alcohol problems during the COVID-19 pandemic:
Insights from a mixed-methods national survey of adults in the United States 

Dear Dr. Gilbert:

I'm pleased to inform you that your manuscript has been deemed suitable for publication in PLOS ONE. Congratulations! Your manuscript is now with our production department. 

Kind regards, 

on behalf of

Dr. Asrat Genet Amnie 

Academic Editor

PLOS ONE